# Tensile and Impact Toughness Properties of a Zr-Based Bulk Metallic Glass Fabricated via Laser Powder-Bed Fusion

**DOI:** 10.3390/ma14195627

**Published:** 2021-09-27

**Authors:** Navid Sohrabi, Annapaola Parrilli, Jamasp Jhabvala, Antonia Neels, Roland E. Logé

**Affiliations:** 1Thermomechanical Metallurgy Laboratory, PX Group Chair, Ecole Polytechnique Fédérale de Lausanne (EPFL), 2002 Neuchâtel, Switzerland; jamasp.jhabvala@epfl.ch (J.J.); roland.loge@epfl.ch (R.E.L.); 2Center for X-ray Analytics, Swiss Federal Laboratories for Materials Science and Technology (Empa), Überlandstrasse 129, 8600 Dübendorf, Switzerland; annapaola.parrilli@empa.ch (A.P.); antonia.neels@empa.ch (A.N.)

**Keywords:** bulk metallic glass (BMG), laser powder-bed fusion (LPBF), crystallization, defects, tensile properties, impact toughness

## Abstract

In the past few years, laser powder-bed fusion (LPBF) of bulk metallic glasses (BMGs) has gained significant interest because of the high heating and cooling rates inherent to the process, providing the means to bypass the crystallization threshold. In this study, (for the first time) the tensile and Charpy impact toughness properties of a Zr-based BMG fabricated via LPBF were investigated. The presence of defects and lack of fusion (LoF) in the near-surface region of the samples resulted in low properties. Increasing the laser power at the borders mitigated LoF formation in the near-surface region, leading to an almost 27% increase in tensile yield strength and impact toughness. Comparatively, increasing the core laser power did not have a significant influence. It was therefore confirmed that, for BMGs like for crystalline alloys, near-surface LoFs are more detrimental than core LoFs. Although increasing the border and core laser power resulted in a higher crystallized fraction, detrimental to the mechanical properties, reducing the formation of LoF defects (confirmed using micro-computed tomography, Micro-CT) was comparatively more important.

## 1. Introduction

Bulk metallic glasses (BMGs) have gained tremendous attention during the last two decades due to their exceptional properties, such as high strength and elastic limit, good corrosion, and wear resistance. However, the main drawback that prevents widespread use of BMGs are their low ductility and size limitation [1].

In BMGs, plastic deformation is confined in shear bands with a nanoscale width, which is then followed by rapid propagation, and finally abrupt rupture [2]. Szuecs et al. [3] improved the low ductility of a Zr-based BMG by introducing a soft crystalline phase and creating a BMG composite with 5.5% ductility in tension. Ning et al. [4] introduced a new approach for achieving ductility in tension without losing the strength of a CuZr-based BMG composite significantly: a B2-CuZr phase formed in the BMG composite and led to deformation-induced martensitic transformation, resulting in 7% ductility.

BMGs have a very low (<2 J) impact toughness [5,6,7,8,9,10,11]. Nagendra et al. [5] reported 90% decrease in the impact toughness of a La-based BMG upon partial crystallization. Roberts et al. [11] could increase the impact toughness of a Zr-based BMG from 0.82 J to 2.66 J by introducing a 67 vol% ductile crystalline phase. Several studies showed that annealing below the glass transition temperature (*T*_g_) of the alloy deteriorates the impact toughness of the BMG due to relaxation and annihilation of the free volume [6,7,8,9]. Another factor reported to be detrimental to the toughness (impact and fracture) of BMGs is the oxygen content [12,13,14].

Although there are some well-established techniques to fabricate BMGs, such as casting and melt spinning, they are not applicable for the fabrication of large size BMGs sensitive to crystallization. For instance, Bordeenithikasem et al. [15] failed to cast large parts from an industrial-grade (high oxygen content) Zr-based metallic glass called AMZ4. They could fabricate large parts with acceptable mechanical properties using laser powder-bed fusion (LPBF). LPBF is the most studied laser-based additive manufacturing (AM) process [16,17]. It leads to extremely high cooling rates (10^4^–10^6^ K/s) due to local and short interaction between the laser and the powder [18]; these are higher than most BMG’s critical cooling rates [19]. Therefore, LPBF provides an excellent opportunity to overcome the size limitation of traditional BMG fabrication methods. Different BMGs based on Pd [20], Al [21], Cu [22], Ti [23], Fe [24,25,26], and Zr [27,28,29,30,31,32] have been processed via LPBF during the last eight years.

Reported works on LPBF of BMG have mainly focused on microstructural evaluation [27,29], crystallization mechanisms [26,27,29], parameters optimization [25,29], and mechanical properties [27,29,30] such as compressive and flexural strength, hardness, and wear resistance. Only one study, Best et al. [33], investigated the tensile properties of a Zr-based BMG fabricated via LPBF. They blamed the presence of defects such as lack of fusion (LoF) for the calamitous failure. They did not increase the input energy, which could decrease the formation of LoFs, to prevent crystallization. Tensile properties of BMGs, and BMG composites fabricated with another laser-based AM process, i.e., direct energy deposition (DED) [34,35,36,37] have been previously investigated. However, to the best of our knowledge, the influence of defects on the impact toughness of BMGs fabricated via AM has not been reported so far.

In the present work, we investigate the tensile properties and impact toughness of a Zr-based BMG (ZrCuAlNb, free of Ni and Be, called AMZ4) produced via LPBF. LoFs are the main reason for the premature failure of the specimens. Two strategies are investigated for changing laser power at the border/contour and in the core of the sample, with the aim of increasing the tensile strength and impact toughness. The effect of border power is shown to be more significant than core parameters. Although the two strategies result in increased detrimental crystallization in AMZ4, the elimination/mitigation of defects remains the key element for property improvement.

## 2. Materials and Methods

In this work, an industrial-grade Zr-based metallic glass (AMZ4) powder with a nominal chemical composition of Zr_59.3_Cu_28.8_Al_10.4_Nb_1.5_ (at.%) and particle size distribution ranging from 10 μm to 50 μm (D_50_ = 30 μm), supplied by Heraeus GmbH was used. A TruPrint 1000 machine (φ 30 μm) was used for the fabrication of the samples in an Ar atmosphere. The processing parameters used for the fabrication of tensile and impact specimens on an Al-Mg-Si1 substrate are mentioned in Table 1 and Figure 1a defines the notions of border/contour, core, and hatching distance. An island scanning strategy with an island size of $4\times 4\;{\mathrm{mm}}^{2}$ and a 90° orientation change after each layer was used for printing specimens for tensile and impact tests.

The geometries of the fabricated specimens (tensile, based on ASTM E8, and impact, based on ISO 14556) are given in Figure 1b. For each Set described in Table 1, three tensile specimens with a thickness of 2.5 mm and six impact specimens were printed. Figure 1c shows one tensile and one impact specimen fabricated via LPBF. The notch of the impact specimens was designed in the CAD file introduced to the LPBF machine. The notch radius was measured as 0.11 mm in the fabricated specimens, slightly larger than the nominal value (0.10 mm). The border region is shown in red color in Figure 1c, and the notch of the impact specimens is surrounded by material scanned with border parameters (in each layer, through the fabrication process). The specimens were milled from one side to remove the support structure; they were then tested in “semi-as-built” condition.

To study the microstructure and fractography, a FESEM (ZEISS GeminiSEM450) was utilized. X-ray diffraction (XRD), using a PanAlytical Empyrean diffractometer with Cu-Κα radiation, was employed to check the amorphous nature of fabricated parts. It was performed on the X-Y cross-section of the printed sample around 0.2 mm below the top surface, which is ten times the LPBF layer thickness (20 μm). At this depth, the material has been subjected to several thermal cycles, similar to those experienced in the bulk. The cross-section includes both core and border regions. Differential scanning calorimetry (DSC) tests were carried out in a Netzsch DSC 204 F1 Phoenix system using a heating rate of 20 °C/min. To etch and reveal the microstructure of the printed specimens, they were cut, ground, and then polished with a suspension of alumina down to 1 μm size and a solution of (45 mL water + 45 mL HNO_3_ + 10 mL HF) was used at room temperature. Figure 1d shows the sample geometry and indicates the locations of SEM and DSC microstructure analyses.

Tensile tests were conducted in a Shimadzu AGS-20kNXD and a Schenck-Trebel universal tester, using a displacement rate of 0.5 mm/min and a DSES-1000 Extensometer. Charpy impact tests (Charpy V-notch tests) were performed using an Amsler machine with a maximum energy of 10 J. Both tests were performed at room temperature.

Micro-computed tomography (Micro-CT) analysis was carried out using an EasyTom XL Ultra 230–160 micro/nano-CT scanner (RX Solutions, Chavanod France). For more information concerning the Micro-CT analysis, please refer to [32]. To determine and select LoFs, the shape parameter was calculated in the Avizo software using the following formula:$$Shape=\frac{S^{3}}{36\pi V^{2}}$$
where *S* is the 3D surface area of the pores and *V* is the volume of the pores. The shape of 1 is considered a perfect sphere, and an increasing shape number leads to more complex convolute shapes [38]. Therefore, a *shape* ≥ 5 was considered as the threshold for detecting LoFs.

## 3. Results

### 3.1. Microstructure

The parameters used for Set A, taken from Ref [27], led to a good combination of density and amorphous content, demonstrated by the resulting high compression and flexural strengths, and high wear resistance. The reason for using a border/contour scan was to improve the geometrical accuracy, as routinely done in LPBF [27,39].

The XRD patterns of printed specimens with the parameters mentioned in Table 1 are presented in Figure 2a. The Set A pattern shows a broad halo peak, which indicates an amorphous structure within the detection limit of XRD. The XRD pattern of Set B shows tiny peaks, which indicates that the amount of crystallization reached a certain level, detectable by XRD. The intensity of crystalline peaks for Set C in Figure 2a is even higher than for Set B. Analyzing the diffraction patterns, we observe overlapping peak domains of intermetallic phases and oxides. The phase analysis has been approached based on the peak indexing and following matching with the COD database. Samples B and C show, in addition to the amorphous hump originating from the amorphous part of the sample, the same diffraction peaks, hence the same crystalline phases. The dominant crystalline phase is Al_2_Zr (PDF No. 99-007-0065, orthorhombic Fdd2) followed by ZrCu (PDF No. 99-007-0025, monoclinic P2_1_/m) and CuZr_2_ (PDF No. 99-007-0022, 96-431-0035, tetragonal I4/mmm). The latter mentioned B2-type CuZr phases in precipitations of CuZr-based BMGs are seen as important for achieving high plasticity CuZr-based BMG composites [40]. Additionally, also Cu_10_Zr_7_ (PDF No. 99-007-0147, orthorhombic Aba2) and Al_2_Zr_3_ (PDF No. 99-007-0063, tetragonal P4_2_/mnm) could crystallize as quoted by other researchers previously [41]. In addition, as the starting powder material showed a high oxygen content, CuO (PDF No. 96-901-5842, monoclinic C2/c) and Cu_2_Zr_4_O can be identified [27,42]. Cu_2_Zr_4_O is a NiTi_2_-type phase exhibiting a cubic cell setting with space group Fd-3m with a = 11.953–12.088 Å. However, the overlap of the different phases and the possible formation of solid solutions do not allow an unambiguous phase attribution.

DSC tests were performed to measure the amount of crystallization in the samples. The DSC curves of Set A, B and C samples are shown in Figure 2b, and quantified in Table 2. To calculate the amorphous fraction, the enthalpy of crystallization (Δ*H*) of the virgin powder, 86.9 J/g in Ref [43], was used as a reference. The onset temperature of glass transition (*T*_g_) and crystallization (*T*_x_) of all three Sets is more or less the same, but the enthalpy of crystallization (Δ*H*) of the samples is different. Comparing Δ*H* to the reference value of the powder, the estimated amorphous fractions are 0.94, 0.91, and 0.83 for Set A, Set B, and Set C, respectively.

Figure 2c shows a back-scattered electron (BSE) image of the X-Z cross-section of Set A. The melt pool and HAZ are clearly detectable. Crystals with darker contrast are dispersed in the amorphous matrix of the HAZ (the light-colored rectangular features are due to contamination.). A higher magnification image (Figure 2d) shows that the crystals size is smaller than 500 nm; they can be considered as nanocrystals. Appendix A shows BSE images of the X-Z cross-section of Set B, in the core region. The width of the HAZ is similar to those of Set A (see Appendix A), since core printing parameters are the same. However, they are more aggregated in the border region of Set B (see Appendix A) because of the increased power in this zone. Appendix A–h shows the melt pool of the core and border regions of Set C, respectively. As expected, in the core region, the width of the HAZ is larger and the nanocrystals are aggregated, due to the increased power. In the border region, the features are almost similar to the ones of Set B (see Appendix A).

### 3.2. Impact Toughness

The direction of the impact force was perpendicular to the building direction of the fabricated specimens. The impact toughness of Set A, Set B, and Set C are measured as 123±28 mJ, 158±19 mJ, and 163±21 mJ, respectively. Set B and Set C showed better toughness than Set A, i.e., the strategy of changing the processing parameters was successful. The statistical scatter of the results is around 12%.

The macrograph of the fracture surface of Set A is presented in Figure 3a, in which LoFs are indicated with yellow-dashed arrows. The systematic presence of LoFs is clearly visible in the near-surface region (see Figure 3b), which corresponds to the intersection of border and core regions (see powder particles trapped in the region between core and border regions in the schematic of Figure 1a). It can be concluded that the border laser power was not high enough to fully melt the powder in that region and resulted in LoF formation. Two easily detectable LoFs were detected in the core of Set B (see Figure 3c) because it has the same parameters as Set A (see Figure 3a). Increasing the border power (Set B) shows promising results in the reduction in the number and size of near-surface LoFs (see Figure 3d) compared to Set A. The fracture surface of Set C in Figure 3e displays no presence of LoFs in the near-surface and core regions. It, therefore, indicates that the increase of core power effectively prevented the formation of LoFs in the core region. However, an open-to-surface porosity (Figure 3f) could be one of the initiation sites of the failure. The jagged morphology (close to the notch root) is detected on the fracture surface of Set B and Set C, Figure 3c and Figure 3e, respectively. The larger jagged region corresponds to the higher toughness (Set B and Set C > Set A). This result is consistent with the finding of Nagendra et al. [5] for impact toughness measurement of a La-based BMG.

Appendix A illustrates a high magnification image from the border (close to the notch) and core of all three Sets, which are representative of the entire fracture surface. Vein-like patterns are evident on the fracture surface, which are attributed to significant softening or reduced viscosity during fracture [44]. However, there is no sign of vein-like patterns in the core of the specimen (2 mm far from the notch into the core), where the morphology corresponds to quasi-brittle fracture (see Appendix A). Unlike ceramics [45] that show very smooth and cleavage fracture surface, dimple-like features are detectable in Appendix A. That is why the fracture mode is here called quasi-brittle. Comparing the near-notch root region for Set A, Set B, and Set C shows that liquid-like features formed in Set B and Set C. These features again correlate with the higher impact toughness and less brittle behavior compared to Set A. The liquid-like features indicate local temperature increase due to the sudden release of stored energy [46].

Micro-CT analyses were performed to investigate and quantify defects inside the specimens in 3D. A part of each specimen (from a reduced section of the tensile specimens and 2 mm away from the notch of the impact specimens) was cut and analyzed, which included both border and core regions. Figure 4 shows 3D constructed (Micro-CT) images from a section of (15.5 mm^3^) impact specimens. The results of the Micro-CT characterization are summarized in Table 3. The overall density of Set A specimens is lower than Set B and Set C, and the number and volume fraction of LoFs is significantly higher for Set A than for Set B and Set C. The two proposed strategies were, therefore, successful in mitigating LoFs. Set B and Set C have similar overall densities, but the number and volume fraction of LoFs is slightly larger for Set B.

### 3.3. Tensile Properties

Tensile tests were carried out with the loading axis perpendicular to the building direction on three specimens for each Set. The tensile yield strength of Set A, B, and C samples are 880±88 MPa, 1120±53 MPa, and 1180±72 MPa, respectively (see Figure 5). The corresponding strains to failure are 1.10±0.15%, 1.38±0.11 , and 1.44±0.14%. Similar to sample A, no macroscopic plastic deformation was detected in Set B and Set C. As can be seen in Figure 6a (Set A), failure starts from the upper-left corner, which is related to the presence of a near-surface LoF (Figure 6b). Figure 6c shows an LoF, which may have formed as a result of a large spatter on the surface. Dimple-like features and vein-like patterns are detected in Figure 6d and Figure 6e, respectively. Figure 6f demonstrates a crack-arrest and shear bands multiplication inside the porosity, which can be due to the blunting of the tip of the crack. While Figure 6d showed a crack arrest by a porosity, Figure 3f indicates instead the initiation of a crack close to a porosity, which is open to the surface. This underlines the effect of size, shape, location, and distribution of porosities in determining their beneficial or detrimental influence on mechanical properties. Figure 6g shows the fracture surface of Set C and the directions of the crack propagation (white arrows). The crack started from the right edge of the specimen. Figure 6h shows a higher magnification of the crack initiation location (region h in Figure 6g). A small near-surface LoF is detected and corresponds to the crack initiation site.

Micro-CT images of the tensile specimens are presented in Appendix A, the summary being presented in Table 3, with trends similar to those of impact samples.

## 4. Discussion

The increased amount of crystallization for Sets B and C compared to Set A is attributed to the higher power used for printing the border (40% higher) and the core (20% higher). As power increases, the input energy increases consequently, and the BMG is more prone to crystallization [31,43]. Although XRD could not detect crystallization for Set A, the increased crystallized fraction measured by DSC from Sets A to C is in agreement with the XRD results. In our previous work [47], using fast scanning calorimetry, we showed that the thermal stability of AMZ4 is low and that the time to crystallization at the nose of the time-temperature-transformation (TTT) diagram is less than 4 ms, which is sufficient for the formation of nanocrystals during the LPBF process.

Deng et al. [48] investigated the effect of density on the compression strength of a CuZr-based BMG composite fabricated via LPBF. They showed that the compression strength increased with an increase in the density of the samples. However, they did not differentiate between the core and near-surface defects. For crystalline materials [49,50], it has been shown that near-surface defects are the most detrimental ones. It is confirmed here that for BMGs as well, such defects, especially those that are in near-surface regions and with a convoluted shape (LoF in Figure 3b), decrease the strength and toughness of the specimen. Only changing the border laser power (comparing Set A and Set B) resulted in a 28% improvement in the impact toughness, while only changing the core power (comparing Set B and Set C) resulted in a 3% improvement. The 3% increase is not significant considering the statistical scatter of the results. The priority is then in removing sub-surface defects, in agreement with Ref. [50]. The number of LoFs (Table 3) is not highly affected as the core power increases (comparing Set B and Set C), but their size is reduced, which lowers the volume fraction of the LoFs (see Figure 4). However, higher core power increases the amount of ‘spherical’ porosities (not considered as LoFs), which explains the very similar densities of Set B and Set C samples, in agreement with Ref [48].

It had been previously shown that using higher laser power leads to an increased amount of crystallization, with a detrimental effect on resistance to cracking [31]. Bordeenithikasem et al. [15] showed that crystallization of AMZ4 resulted in the increase of brittleness and reduction of flexural strength. In addition, Pacheco et al. [41] and Marattukalam et al. [43] detected crystallization in AMZ4. However, there is no report on the formation of ductile crystals during the crystallization of AMZ4 in the literature, which could improve the ductility of the fabricated parts. The present study shows, however, that removing defects (mainly LoFs) and increasing crystallization does improve the impact toughness, overall. This assertion is valid for specimens with crystallized fractions up to 17% (Set C). Higher levels of crystallization have not been investigated in the current study but are reported in Appendix A from other studies.

There is no literature study on the impact toughness of either BMGs fabricated via AM methods or AMZ4 fabricated by other conventional methods such as casting. The latter would be of interest to establish reference properties to which our LPBF BMGs could be compared. The results of this study are therefore compared with those from other BMGs fabricated by casting and semi-solid forged, in Appendix A. Our results indicate values lower than those of amorphous Zr-based BMGs [7,8,9,10,11] and Al-alloys [10], but close to those of Zr-based annealed samples in Refs [7,8,9]. The notch in the as-cast samples was prepared by precise machining, whereas in the current study, it was included in the LPBF fabrication. The notch curvature of the specimens was slightly higher than the nominal value due to the limited resolution of the LPBF process and might overestimate the absolute values of measured impact toughness. However, the presence of the roughness inherent to the LPBF process in the as-built condition could result in stress concentration, which would then lead to underestimating the impact toughness results. It is understood that comparing the absolute value of impact toughness for specimens with different geometries, test methods, and fabrication methods cannot provide a clear ranking as long as the values are in the same order of magnitude. Some of the BMG composites containing a ductile crystalline phase [10,11] showed enhanced impact toughness, even higher than that of Ti-6Al-4V [10]. In general, however, unwanted crystallization [5] and annealing [6,7,8,9] dramatically reduced the impact toughness.

Best et al. [33] studied the effect of the oxygen content in BMGs on fracture toughness. Although lab-grade (~170 ppm oxygen content) and industrial-grade (~1300 ppm oxygen content) samples had similar hardness and compressive strength, the lab-grade sample had a five times higher fracture toughness. The authors correlated this behavior to the dissolved oxygen content, which acts as a barrier for atomic movement and reduces toughness. Yokoyama et al. [51] also commented on the deleterious effect of oxygen on the impact toughness of a Zr-based BMG. The oxygen content of LPBF parts was measured as 1480 ppm in our previous work [27]. According to Lin et al. [36], annealing was the reason for the absence of ductility in a compression test of a BMG fabricated via DED. A combination of partial crystallization, high oxygen content, annealing, and the presence of printing defects all potentially explain the low measured values of impact toughness in the present study.

No macroscopic plastic deformation was detected in the tensile tests (Figure 5), in agreement with Refs [35,36]. Lin et al. [36] correlated this behavior to the annealing effect, which happens in laser-based AM processes because of the associated cyclic heating. The formation of dimples is caused by the nano-scaled cavitation in the quasi-brittle fracture [52,53]. The change of features morphology on the fracture surface from vein-like patterns to dimples correspond to a transition of the fracture mode from ductile to quasi-brittle [54].

Since there is only one reported study [33] on the tensile properties of BMGs fabricated via LPBF, the present results were also compared to properties measured on BMGs fabricated via other AM methods, such as DED [34,35,36,37]. Figure 7 plots our AMZ4 results together with those of Ref. [33], and four other Zr-based BMGs [34,35,36,37]. It is assumed that in Ref. [33], the tensile samples were ground before testing since no printing pattern can be observed on the external surface. The reason for the slightly lower tensile strength in Ref. [33] compared to Set B and Set C can be attributed to the presence of large and irregular porosities (LoFs) in the printed samples, which Best et al. [33] showed by providing a Micro-CT image. Although our AMZ4 specimens were in “semi-as-built” condition (without any surface finishing), whereas the tensile specimens in Refs [34,35,36,37] were in a surface finished state, the present results position themselves among the best ones for BMGs fabricated via AM processes. Therefore, AMZ4 manufactured via LPBF can be a good candidate for applications where a high tensile yield strength is required.

LoFs were present in both impact and tensile specimens, and the near-surface LoFs were identified as the primary reason for premature failure. This is in agreement with the finding of Best et al. [33] and Su et al. [34], who showed that printing defects resulted in low tensile strength of BMGs fabricated via AM processes. In addition, near-surface defects are also known as “critical crack initiation pores” [49,50]. Going from sample A to B leads to a 27% of improvement in the yield strength, while sample C improves only by 5% compared to Set B (see Figure 7). Due to the statistical scatter of the tensile results around 5%, the improvement is not considered to be significant. Both tensile and impact toughness results show the importance of eliminating/mitigating the near-surface defects, in particular the LoFs.

## 5. Conclusions

In this work, crack-free Zr-based bulk metallic glass (BMG) specimens were fabricated via laser powder-bed fusion (LPBF). Tensile and impact toughness properties of the fabricated specimens were investigated in the “semi-as-built” state. The following conclusions could be reached:Using the first strategy (increasing the laser power in the border region) led to the mitigation of near-surface LoFs, and the results showed around 27% improvement in the tensile yield strength and impact toughness. Changing only the core laser power (second strategy) resulted in only 5% and 3% improvement in the tensile yield strength and impact toughness, respectively, compared to the first strategy, which are not considered to be significant.It was confirmed that, for additively manufactured BMGs, near-surface defects such as lack of fusions (LoFs) are more critical than those present in the core of the specimens.Although implementing the two mentioned strategies increased crystallization (up to 17%) because of the higher laser power, the importance of reducing porosity defects (especially LoFs) prevailed. The tensile yield strength of the Zr-based BMG was one of the best ones among those reported for BMGs fabricated via additive manufacturing (AM).

## Figures and Tables

**Figure 1 materials-14-05627-f001:**
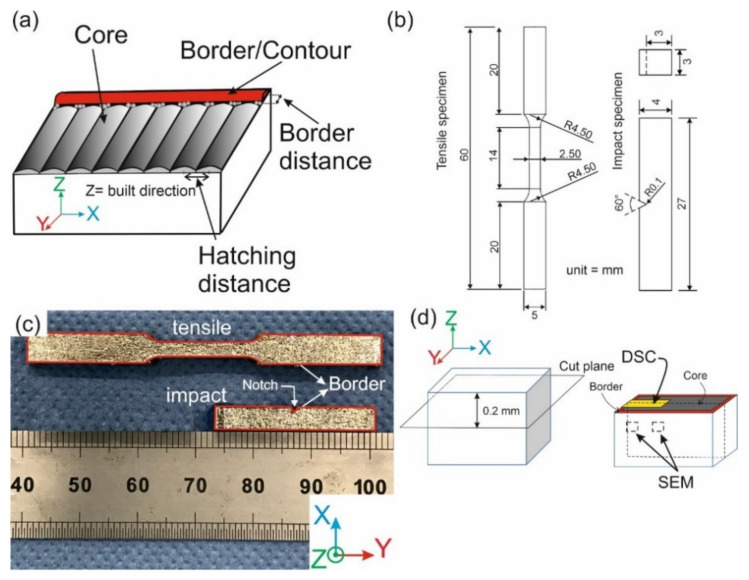
(**a**) Schematic view of laser scanning showing the notions of border, core, hatching distance, and border distance for a part of a sample, (**b**) geometry of the tensile and impact specimens, (**c**) a tensile and an impact specimen fabricated via LPBF, and (**d**) schematic view of the samples showing locations of SEM and DSC analyses with respect to the core and border regions.

**Figure 2 materials-14-05627-f002:**
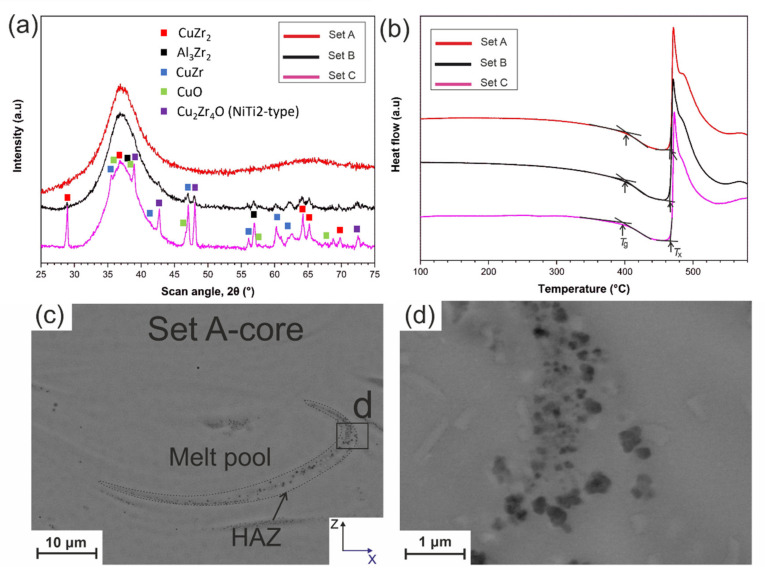
(**a**) XRD patterns of Sets A–C. (**b**) DSC results of the samples manufactured by Sets A–C parameters using a heating rate of 20 °C/min, (**c**) back-scattered electron (BSE) image of the X-Z cross-section of Set A, and (**d**) higher magnification of region d in (**c**) indicating the presence of nanocrystals. The light-colored rectangular features are due to contamination.

**Figure 3 materials-14-05627-f003:**
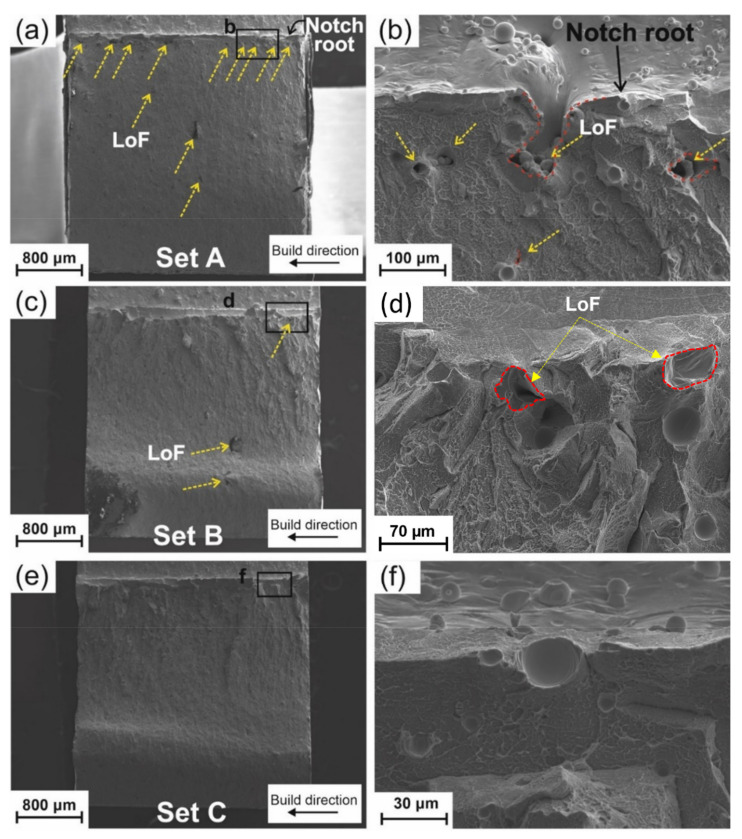
(**a**) Fracture surface of a Set A impact specimen, (**b**) higher magnification image of region b in (**a**), (**c**) fracture surface of a Set B impact specimen, (**d**) higher magnification image of region d in (**c**), (**e**) fracture surface of a Set C impact specimen, and (**f**) higher magnification image of region f in (**e**), shear bands were formed inside the large porosity.

**Figure 4 materials-14-05627-f004:**
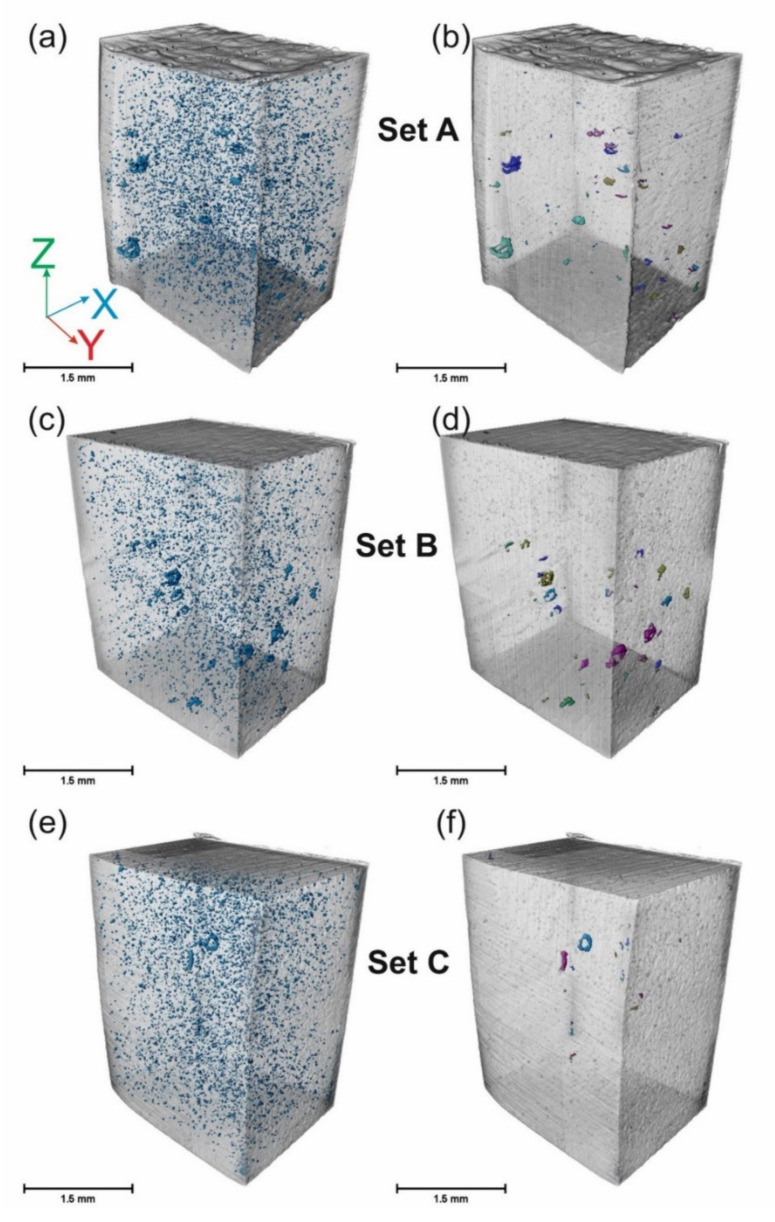
A 3D constructed image of a part of impact specimens (15.5 mm^3^) fabricated via (**a**,**b**) Set A, (**c**,**d**) Set B, and (**e**,**f**) Set C, using X-ray micro-computed tomography (Micro-CT). Specimens (**a**,**c**,**e**) include all defects, while (**b**,**d**,**f**) isolate LoFs. All the samples contained border and core regions.

**Figure 5 materials-14-05627-f005:**
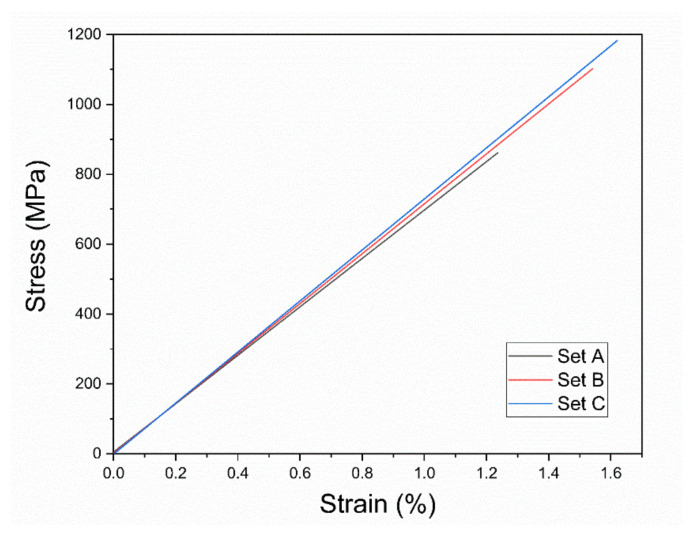
Stress-strain curve of tensile tests for Sets A–C.

**Figure 6 materials-14-05627-f006:**
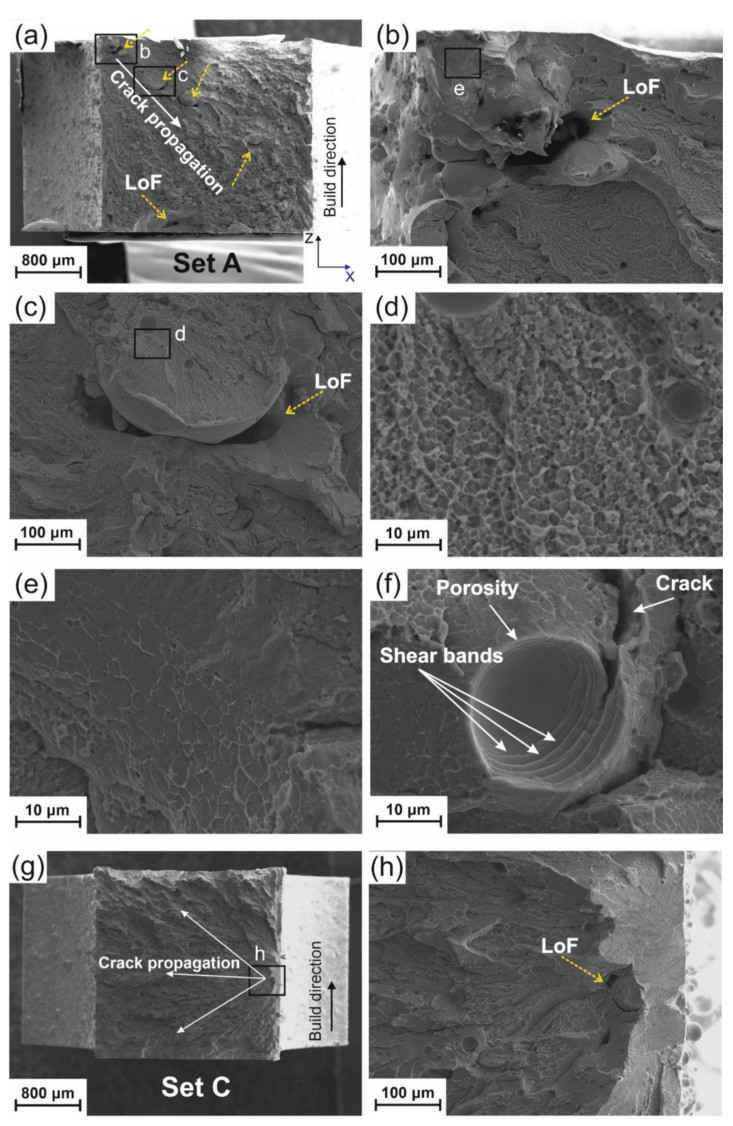
(**a**) Fracture surface (X-Z plane) of a tensile specimen Set A after rupture. Yellow-dashed arrows show LoFs. (**b**) Region b in (**a**) with a higher magnification showing the start of failure from a lack of fusion (LoF), (**c**) region c in (**a**) with a higher magnification indicating another LoF, (**d**) region d in (**c**) with a higher magnification showing dimple-like features, (**e**) region e in (**b**) with a higher magnification indicating vein-like patterns, (**f**) a porosity arrests a crack and multiple shear bands are found, (**g**) fracture surface (X-Z plane) of a tensile specimen C, and (**h**) region h shown in (**g**) with a higher magnification.

**Figure 7 materials-14-05627-f007:**
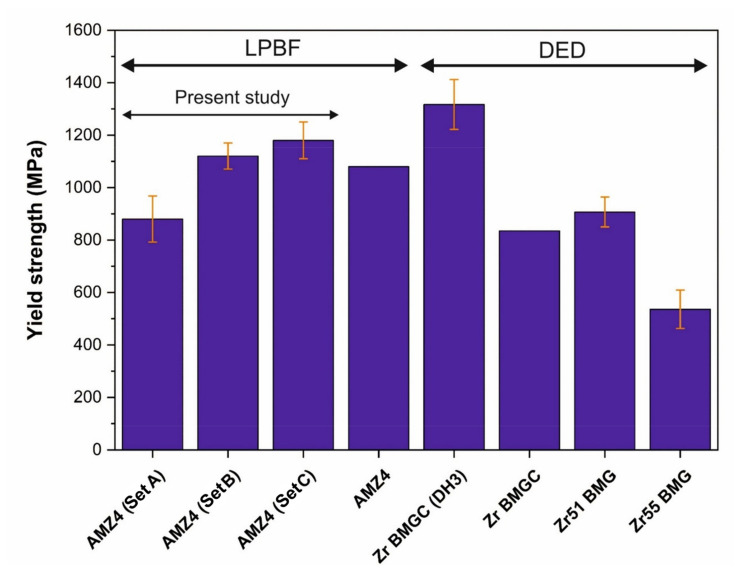
Comparison of tensile yield strengths in the present study with a LPBF manufactured AMZ4 [33], and with DH3 (Zr_39.6_Ti_33.9_Nb_7.6_Cu_6.4_Be_12.5_) [37], Zr BMG composite (Zr_39.6_Ti_33.9_Nb_7.6_Cu_6.4_Be_12.5_) [34], Zr51 BMG (Zr_51_Ti_5_Ni_10_Cu_25_Al_9_) [35], and Zr55 BMG (Zr_55_Cu_30_Al_10_Ni_5_) [36] fabricated via direct energy deposition (DED).

**Table 1 materials-14-05627-t001:** Printing parameters used for the fabrication of tensile and impact specimens. The notions of “core” and “border” are illustrated in Figure 1a.

**Set**	**Core Power (W)**	**Scanning Speed (mm/s)**	**Hatching Distance (μm)**	**Layer Thickness (μm)**	**Border Power (W)**
A	30	600	90	20	30
B	30	600	90	20	42
C	36	600	90	20	42

**Table 2 materials-14-05627-t002:** DSC results (using a heating rate of 20 °C/min) of samples manufactured with Sets A-C parameters. The amorphous fraction is calculated by dividing the enthalpy of crystallization (Δ*H*) of each Set by the Δ*H* value of the virgin powder, which was 86.9 J/g in Ref. [43].

Sample	*T*_g_ (°C)	*T*_x_ (°C)	Δ*H* (J/g)	Amorphous Fraction
Set A	393.7	467.6	81.34	0.94
Set B	394.8	466.7	79.39	0.91
Set C	394.7	468.5	72.69	0.83

**Table 3 materials-14-05627-t003:** Density of specimens (including both border and core regions), number, and volume fraction of LoFs present in the impact and tensile specimens.

Sample	Density (%) *	Number of LoFs	Volume Fraction of LoFs (%)
Impact- Set A	99.80	73	0.051
Impact- Set B	99.85	48	0.023
Impact- Set C	99.86	46	0.007
Tensile- Set A	99.77	129	0.068
Tensile- Set B	99.87	38	0.014
Tensile- Set C	99.87	34	0.009

* Within the resolution of the Micro-CT method.

## Data Availability

The raw/processed data required to reproduce these findings cannot be shared at this time as the data also forms part of an ongoing study.

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
