# Peer review of "Tensile and Impact Toughness Properties of a Zr-Based Bulk Metallic Glass Fabricated via Laser Powder-Bed Fusion"

_materials, 2021, doi:10.3390/ma14195627_

Round 1

Reviewer 1 Report

The manuscript Tensile and impact toughness properties of a Zr-based bulk metallic glass fabricated via laser powder-bed fusion studies tensile and fracture behavior of LPBF fabricated BMG. The experimental results are reported in detail and well presented. In general, the review thinks the manuscript is suitable for publication. However, the following comments should be considered.

  1. The manuscript focus on the effect of LoFs. However, other features are also obvious visible on the fracture surface, including “crystallization” and “porosity”. Their effects on fracture toughness haven’t been discussed.

  1. Although Micro CT has been reported in the literature (the author also cited), some useful explanation about how to distinguish LoF and other defects is necessary.

  1. Line 14-15,”The presence of lack of fusion (LoF) defects in the near-surface region of the samples resulted in low properties.” This sentence reads very confusingly for the first time. Maybe it should start with “The presence of the defect which is lack of fusion (LoF)……”

  1. Figure 2b, The Y axis title “heat flow” is hidden

  1. The reviewer suggests that the authors combine the manuscript with the supplementary information for the convenience of the readers.

Reviewer 2 Report

See attachment.

Reviewer 3 Report

Sohrabi et al studied tensile and impact toughness properties of a Zr-based bulk metallic glass fabricated via laser powder-bed fusion method. The experimental results and related findings have important reference significance for the preparation of bulk metallic glass with high mechanical properties by the same processing technology. The work is interesting and has a strong appeal to the readers. The paper can be considered for acceptance after a minor revision.

1. Page 4. line 132.
Although studying the formed phase/s is not the focus of the present study, these formed phases should be identified through comparing with PDF(ICPDS) cards. Because these crystalline phases possess crystallized fractions up to 17%, they may be an important factor affecting the mechanical properties of materials. They should be reported positively for further study by other researchers. 

2. The conclusion section is too long. It is suggested that the author reorganize and only keep the most important conclusions.

3. Some references are too old, it is suggested to cite more literatures in the past two years. In addition, the writing format of some documents is not correct. For example, the numbers in the material components should be subscript.

4. Three-line tables are required. Some pictures and titles are also incorrectly formatted. For details, see the template format.

Round 2

Reviewer 1 Report

No further questions from the reviewer.

Reviewer 2 Report

Comments are fully addressed by authors.